

# Genome-wide identification and characterization of the *AP2/ERF* gene family in loblolly pine (*Pinus taeda* L.)

Peiqi Ye[1,2], Xiaoliang Che[2], Yang Liu[2], Ming Zeng[2], Wenbing Guo[2], Yongbin Long[2], Tianyi Liu[1] and Zhe Wang[2]

[1] Guangdong Key Laboratory for Innovative Development and Utilization of Forest Plant Germplasm, South China Agricultural University, Guangzhou, Guangdong, China
[2] Guangdong Provincial Key Laboratory of Silviculture, Protection and Utilization, Guangdong Academy of Forestry, Guangzhou, Guangdong, China

## ABSTRACT

The loblolly pine (*Pinus taeda* L.) is one of the most profitable forest species worldwide owing to its quick growth, high wood yields, and strong adaptability. The *AP2/ERF* gene family plays a widespread role in the physiological processes of plant defense responses and the biosynthesis of metabolites. Nevertheless, there are no reports on this gene family in loblolly pine (*P. taeda*). In this study, a total of 303 members of the *AP2/ERF* gene family were identified. Through multiple sequence alignment and phylogenetic analysis, they were classified into four subfamilies, including AP2 (34), RAV (17), ERF (251), and Soloist (1). An analysis of the conservation domains, conserved motifs, and gene structure revealed that every PtAP2/ERF transcription factor (TF) had at least one AP2 domain. While evolutionary conservation was displayed within the same subfamilies, the distribution of conserved domains, conserved motifs, and gene architectures varied between subfamilies. Cis-element analysis revealed abundant light-responsive elements, phytohormone-responsive elements, and stress-responsive elements in the promoter of the *PtAP2/ERF* genes. Gene Ontology (GO) and Kyoto Encyclopedia of Genes and Genomes (KEGG) analyses of potential target genes showed that the *AP2/ERF* gene family might play a critical role in plant growth and development, the response to environmental stresses, and metabolite biosynthesis. Utilizing quantitative real-time PCR (qRT-PCR), we examined the expression patterns of 10 randomly selected genes from Group IX after 6 h of treatments with mechanical injury, ethephon (Eth), and methyl jasmonate (MeJA). The *AP2/ERF* gene family in the loblolly pine was systematically analyzed for the first time in this study, offering a theoretical basis for exploring the functions and applications of *AP2/ERF* genes.

## INTRODUCTION

To control the expression of downstream functional genes, transcription factors (TFs) can precisely bind to cis-acting elements in their promoter regions. These interactions play a crucial regulatory role in the way that plants respond to stress, as well as in the regulation

Corresponding authors
Tianyi Liu, tianyiliu@scau.edu.cn
Zhe Wang, wangzhe@sinogaf.cn

of plant growth and development (*Singh, Foley & Oñate-Sánchez, 2002*; *Hernandez-Garcia & Finer, 2014*). One of the most significant groups of TFs in plants is the AP2/ERF family. It was initially identified in *Arabidopsis thaliana*; subsequently, numerous homologous proteins with AP2 domains have been discovered in other plants, including tobacco, maize, and others (*Jofuku et al., 1994*; *Ohme-Takagi & Shinshi, 1995*; *Zhuang et al., 2010*; *Licausi, Ohme-Takagi & Perata, 2013*). The structural characteristics of the *AP2/ERF* gene family encompass the presence of at least one conserved AP2/ERF domain, which consists of 58 to 70 amino acid residues arranged in three antiparallel β-sheets and one α-helix (*Okamuro et al., 1997*; *Allen et al., 1998*). *Sakuma et al. (2002)* classified AP2/ERF TFs into five subfamilies based on the number and similarity of DNA-binding domains: the AP2 (APETALA2), ERF (ethylene-responsive element-binding factor), DREB/CBF (dehydration-responsive element/C-repeat), RAV (related to AB13/VP) and Soloist (few unclassified factors). The AP2 family members contain one or two complete AP2 domains, which are associated with plant seed regulation and the development of floral organs (*Luo et al., 2021*; *Zumajo-Cardona, Pabon-Mora & Ambrose, 2021*). In the ERF subfamily, which may bind to the GCC-box element in particular and is implicated in the regulation of the plant response to ethylene and abiotic stresses, alanine (A) and aspartic acid (D) occupy the 14th and 19th positions of its AP2 structural domain, respectively (*Sakuma et al., 2002*). The DREB subfamily, with valine (V) at position 14 and glutamate (E) at position 19 of the AP2 domains, binds specifically to DRE/CRT elements and is mainly involved in the regulation of processes such as the response to abscisic acid (ABA), drought, and low-temperature (*Ohme-Takagi & Shinshi, 1995*). *Nakano et al. (2006)* combined the DREB and ERF genes into ERF based on conserved structural domain features, gene structure, and additional genome annotation information in *A. thaliana*, and classified them into four subfamilies: AP2, ERF (Groups I–X, VI–L, and Xb–L), RAV, and Soloist. The results of these two classification methods are essentially consistent. The RAV subfamily, which consists of a single AP2 domain along with a B3 domain, is mostly linked to the phytohormone response and abiotic stress response (*Zhao et al., 2017*). Soloist also contains one AP2 domain, the sequence of which differs significantly from that of the other subfamilies but is more conserved between species.

With the advent of high-throughput sequencing, genetic resources like genomes and transcriptomes are now more plentiful. Existing research findings indicate that the *AP2/ERF* gene family is one of the largest transcription factor families. In model plants, the *AP2/ERF* gene family has 147, 170, and 375 members in *A. thaliana*, *Oryza sativa*, and *Nicotiana tabacum*, respectively (*Nakano et al., 2006*; *Rashid et al., 2012*; *Gao et al., 2020*). In herbaceous plants, the *AP2/ERF* gene family has 214, 120, and 138 members in *Daucus carota*, *Dendrobium catenatum*, and *Boehmeria nivea*, respectively (*Li et al., 2015*; *Han et al., 2022*; *Qiu et al., 2022*). In woody plants, the *AP2/ERF* gene family has 172, 234, and 364 members in *Elaeis guineensis*, *Pyrus pyrifolia*, and *Salix matsudana*, respectively (*Zhou & Yarra, 2021*; *Xu et al., 2023*; *Zhang et al., 2021*). In most species, the AP2/ERF family is abundant and plays a crucial regulatory role in various physiological processes of plants. Firstly, AP2/ERF TFs play a role in the growth and development of plants. PtaERF003 is responsible for enhancing the development of adventitious roots and promoting the

proliferation of lateral roots in *Populus* (*Trupiano et al., 2013*). In *Brassica rapa*, BrAP2 assumes a pivotal function in the modification of sepals, shaping their unique characteristics (*Zhang et al., 2018*). Secondly, AP2/ERF TFs are involved in the secondary metabolism of plants. FtERF-EAR3 interacts with *FtF3H* in *Fagopyrum tataricum* to suppress the expression of *FtF3H* and the biosynthesis of flavonoids (*Ding et al., 2022*). LbERF5.1 interacts with *CCD4.1* in *Lycium barbarum* to regulate the accumulation of carotenoids (*Zhao et al., 2023*). Thirdly, AP2/ERF TFs influence the stress resistance of plants. While OsBIERF3 plays a positive role in bolstering immunity against fungi and bacteria in rice, it paradoxically functions as a negative modulator of cold stress tolerance (*Hong et al., 2022*). Wheat exhibits the induction of TaERF-6-3A in response to several environmental stresses, including drought, salt, and cold (*Yu et al., 2022*). The stress resistance of plants has been a highly researched area in recent years. AP2/ERF TFs are becoming an increasingly frequent topic in research on biotic and abiotic stress (*Nie & Wang, 2023*). Plants, especially perennial woody plants, critically rely on their adaptability to external changes.

Pinaceae species are some of the oldest tree species in the world and have developed a rapid and comprehensive response system to external stimuli throughout their evolutionary history. Loblolly pine, native to the southeastern United States, has become a predominant tree species for cultivation in subtropical regions due to its strong adaptability, rapid growth, straight trunk formation, high-quality wood, and substantial resin production (*Lu et al., 2017*). Its wood can be utilized for pulp, construction, and fiber materials. For humans, resin is a kind of chemical raw material, with applications in fields such as textiles, coatings, rubber, fragrances, and pharmaceuticals (*Correa et al., 2013*). But for the tree organism, the secretion of resin represents the most crucial defense mechanism in response to various stresses (*Vázquez-González et al., 2020*). Resin is primarily composed of terpenoid compounds. Previous studies have shown that AP2/ERF TFs play a role in transcriptional regulation during both stress responses and the biosynthesis of metabolites in many plants. Diterpenoid production in *Salvia miltiorrhiza* was positively regulated by SmERF128 (*Zhang et al., 2019*). LcERF19 elevated the expression of the sesquiterpene synthase gene *LcTPS42*, consequently enhancing the biosynthesis of geraniol and neral in *Litsea cubeba* (*Wang et al., 2022*). These findings suggest that AP2/ERF TFs in loblolly pine may function similarly in the regulation of turpentine, thereby contributing to stress responses. Furthermore, the transcriptome-wide identification and characterization of AP2/ERF genes were conducted in *P. massoniana*, showing that some of them exhibit differential expression under pine wood nematode (PWN) or drought treatment (*Zhu et al., 2021*; *Sun et al., 2022*). Another functional analysis revealed that the overexpression of *PmERF1* could improve plants' ability to tolerate drought (*Zhang et al., 2023*). Transcriptome-based investigations of resin production in *P. massoniana* and *P. taeda* have revealed that specific members of the *AP2/ERF* gene family show differential expression (*Bai et al., 2020a*, *2020b*; *Mao et al., 2021*). These findings proved that the *AP2/ERF* gene family in Pinaceae species plays a transcriptional regulatory role in response to biotic and abiotic stresses. With the completion of the loblolly pine genome assembly and the enhancement of its quality, genetic reservoirs have become accessible for the

comprehensive identification and in-depth functional analysis of the *AP2/ERF* gene family in loblolly pine (*PtAP2/ERF*) ([Neale et al., 2014](); [Zimin et al., 2017]()). Hence, investigating the *AP2/ERF* gene family in loblolly pine is expected to provide a reference for further research on the function of the *PtAP2/ERF* gene family and improve plant resistance from the perspective of the genetic improvement of forests.

The loblolly pine genome version 2.01 was used in this study to identify, categorize, and analyze the AP2/ERF superfamily. The objectives of this study are as follows: (1) to perform comprehensive identification and classification of the *AP2/ERF* gene family in loblolly pine; (2) based on this classification, to analyze the physicochemical properties, conserved domains, conserved motifs, gene structures, and cis-acting elements characteristic of the *AP2/ERF* gene family; (3) in combination with the predicted target genes and the expression profiles of certain ERF members under defense-related treatments, to provide initial insights into the potential roles played by the *AP2/ERF* gene family in loblolly pine.

## MATERIALS AND METHODS

### Genome-wide identification and physicochemical properties of PtAP2/ERF TFs

The loblolly pine genome and its annotations were downloaded from the TreeGenes database ([http://treegenesdb.org/]()). To identify the members of the AP2/ERF family in the genome, a double approach was used in TBtools software ([Chen et al., 2020]()). The protein sequences of *AP2/ERF* genes in *A. thaliana* were downloaded from The *Arabidopsis* Information Resource (TAIR) database ([https://www.arabidopsis.org/]()) and used as query sequences in the BlastP program (E-value $< e^{-5}$). The Pfam database ([http://pfam.xfam.org/]()) provided the Hidden Markov Model (HMM) profiles of AP2 domains (PF00847). This file was subsequently used in a Simple HMM search against the loblolly pine genome in TBtools software (E-value $< e^{-5}$). The sequences obtained were sent to the Simple Modular Architecture Research Tool (SMART) ([http://smart.embl-heidelberg.de]()) and NCBI Conserved Domain Search (CD-Search) ([https://www.ncbi.nlm.nih.gov/Structure/cdd/wrpsb.cgi]()) to confirm the integrity of the AP2 domain.

The ExPASy ProtParam tool ([http://web.expasy.org/protparam/]()) was used to examine the physicochemical characteristics of the PtAP2/ERF proteins, including the number of amino acids, molecular weight, theoretical pI, instability index, and grand average of hydropathicity. WoLF PSORT ([https://wolfpsort.hgc.jp/]()) and DeepTMHMM ([https://dtu.biolib.com/DeepTMHMM]()) were used to carry out subcellular localization and transmembrane structure prediction, respectively.

### Classification and phylogenetic analysis of PtAP2/ERF protein

Multiple sequence alignment of the AP2/ERF proteins was performed using the ClustalW program in MEGA 7 ([Kumar, Stecher & Tamura, 2016]()). The AP2 domain sequences of AP2/ERF proteins from three species—*A. thaliana* ([Nakano et al., 2006]()), *P. massoniana* ([Zhu et al., 2021]()), and *P. taeda*—were used to build a maximum likelihood tree with 1,000 bootstrap repetitions following multiple sequence alignment. Interactive Tree Of Life

(iTOL) (https://itol.embl.de/) was used for the display, annotation, and management of phylogenetic trees (*Letunic & Bork, 2021*). The *PtAP2/ERF* gene family was classified with reference to the *AP2/ERF* gene family in *Arabidopsis* (*Nakano et al., 2006*).

## Conserved domain of analysis of PtAP2/ERF protein

The conserved domains of the PtAP2/ERF proteins were analyzed using the CD-Search tool (https://www.ncbi.nlm.nih.gov/Structure/cdd/wrpsb.cgi). Multiple sequence alignment of the AP2/ERF proteins was performed using the ClustalW program in MEGA 7 (*Neale et al., 2014*). The result of the multiple sequence alignment was visualized using GeneDoc software.

## Conserved motif, gene structure, and promoter analysis of PtAP2/ERFs

The conserved motifs were analyzed using the online MEME program (https://meme-suite.org/meme/tools/meme). The conserved motifs and gene structure were visualized using TBtools software.

The 2.0 kp sequence upstream of the transcriptional start site of the *PtAP2/ERF* genes was submitted to the online site PlantCARE (http://bioinformatics.psb.ugent.be/webtools/plantcare/html/) to predict their cis-acting elements.

## Prediction and analysis of PtAP2/ERF target genes

Target genes with the ERF protein binding site elements DRE/CRT (G/ACCGAC) and GCC-Box (AGCCGCC) were investigated using TBtools. Gene Ontology (GO) analysis and Kyoto Encyclopedia of Genes and Genomes (KEGG) Enrichment analyses of target genes were performed using eggNOG-mapper (http://eggnog-mapper.embl.de/) and visualized using the OmicShare tools (https://www.omicshare.com/tools).

## Plant materials, treatments, and quantitative real-time PCR

Previous studies have shown that ERF-IX of the *AP2/ERF* gene family is closely associated with defense responses and the biosynthesis of secondary metabolites in plants (*Nakano et al., 2006*; *Wang et al., 2022*; *Shoji & Yuan, 2021*). Ethephon (Eth) and methyl jasmonate (MeJA) are both plant hormones capable of effectively activating the defense responses in pine trees; they also serve as active components in stimulant pastes (*Lopez-Alvarez et al., 2023*; *de Oliveira Junkes et al., 2019*). To investigate whether PtERFs play a regulatory role in the defense of loblolly pine, we randomly selected ten genes in Group IX and analyzed the expression of ten selected genes in response to defense-related treatments with mechanical injury, Eth, and MeJA *via* qRT-qPCR. In this study, we selected 36 healthy and uniformly growing one-year-old loblolly pine seedlings from the Yingde Research Institute of Forestry in Guangdong Province, China. These seedlings were randomly divided into two groups: the treated groups and the control groups, with 18 seedlings in each group. The treated groups were further divided into three subgroups, with six seedlings in each subgroup. These subgroups were subjected to different treatments: mechanical injury, spraying with 500 μmol/L Eth and treatment with 10 mol/L MeJA. Mechanical injury involved making wounds on the parts of the plants starting from 15 cm above the ground,

**Table 1 Primer information for qRT-PCR.**

| Gene name | Forward primer (5′-3′) | Reverse primer (5′-3′) |
|---|---|---|
| *Actin* | GAGCAAAGAGATCACTGCACTTG | CTCATATTCGGTCTTGGCAATCC |
| *PITA_03224* | GAGCAGCGTAAAACAAGAAGTC | TCCCGTACTGAAGAACAAGG |
| *PITA_06992* | TCGGAGGACATGGTTTTGTA | GCTGTTCGGTTCTCGTTTCT |
| *PITA_08629* | TTTTCCCCTTGATTTGGTTG | TTTTCTTCTCCCCTCTGTGC |
| *PITA_23504* | CCGAGCAAATCTCAAAAGCC | TGTTAGACGCAACGGAACGT |
| *PITA_24252* | GCAGTCCCGAAGAAGAAAAC | CAGGCAGCAGATAAACAAGG |
| *PITA_25673* | CTTCAGCACTGGCACTGGGTT | GGTAAGTTCGCAGCGGAGAT |
| *PITA_26508* | ATGTCGTCACCGCAGCCTTAG | AGATTTGCCCATCGTCCCAC |
| *PITA_29504* | CCGAACACCCAAAGAACAGC | CCAAACTCTAGCGCCCTTACGCGCA |
| *PITA_42370* | TGAAGGTGCCGAGCCAAACT | GCCGATGGAAACGAAGAAGC |
| *PITA_45859* | CCCCTGTAATGCCTACCAAC | TAACGCACAAGGAGGAGATG |

creating wounds every 4 cm that were 5 mm × 5 mm in size, with approximately 4–5 wounds per plant. Following the mechanical injury treatment, resin flow from the wounds could be observed. The Eth treatment involved spraying the aerial parts of the plants with a solution of 500 μmol/L Eth prepared in distilled water, with approximately 80 ml applied per plant. The MeJA treatment involved spraying the plants with a 10 mol/L MeJA solution prepared in an organic solution containing 0.1% Tween 20 and 0.1% ethanol, applied in a manner similar to the Eth treatment. Each of these treatments corresponded to three control groups: CK1, CK2, and CK3, with six seedlings in each control group. CK1 represented the untreated control group, CK2 underwent the application of distilled water, and CK3 underwent the application of an organic solution containing 0.1% Tween 20 and 0.1% ethanol. After 6 h of treatment, three plants from each group with similar growth conditions were sampled, and freshly collected needles were immediately frozen in liquid nitrogen and stored at −80 °C.

According to the manufacturer's instructions, the total RNA of the needles was extracted using a DP441 RNAprep Pure Kit (Tiangen Biotech, Beijing, China). The concentration and purity of the extracted RNA were assessed using a Biochrom Bio Drop Duo, and its integrity was detected *via* 1% agarose gel electrophoresis. cDNA was synthesized through the reverse transcription of 800 ng of RNA using HiScript III RT SuperMix for qPCR (+gDNA wiper) (Vazemy, Nanjing, China). Quantitative real-time PCR (qRT-PCR) was performed using a BIO-RAD CFX Connect Real-Time system using the ChamQ Universal SYBR qPCR Master Mix (Vazemy, Nanjing, China). A 20 μL reaction containing 1 μL synthesized cDNA, 10 μL 2 × ChamQ Universal SYBR qPCR Master mix, 0.5 μL 10 μM forward primer, 0.5 μL 10 μM reverse primer, and 8 μL ddH$_2$O was amplified. Complete reaction conditions: one cycle of 95 °C for 30 s, followed by 40 cycles of 95 °C for 10 s, 60 °C for 30 s. To standardize the gene expression levels, Actin was chosen as the internal reference gene. The calculation of relative expression levels was carried out using the $2^{-\Delta\Delta Ct}$ method. Primer5 was used for designing specific amplification primers for qRT-PCR (Table 1).
## Statistical analysis

Each experiment consisted of three independent biological replicates. The results were analyzed to determine significance (One-way ANOVA) using SPSS 26 software and visualized using GraphPad Prism 9. The results display the mean and standard error, while the p values indicate the significance of the differences compared to the controls.

# RESULTS

## Identification, physicochemical properties, subcellular localization, and transmembrane structure prediction of PtAP2/ERF TFs

The sequences containing incomplete or absent AP2 domains were removed, and 303 *AP2/ERF* genes were identified in the *P. taeda* genome. The protein sequences and coding sequences of the *AP2/ERF* gene family in loblolly pine are listed in Table S1. Among these genes, 18 proteins contain double AP2 domains, 269 proteins contain a single AP2 domain, 15 proteins contain AP2 domains and a B3 domain, and one protein contains three AP2 domains and a B3 domain.

We further analyzed the fundamental characteristics of these proteins, which included their physicochemical properties, subcellular localization, and transmembrane structure (Table S2). The physicochemical investigation revealed differences in the protein length, molecular weight, theoretical isoelectric point, and instability index among the members of the AP2/ERF family in loblolly pine. The number of amino acids, molecular weight, theoretical pI, and instability index are in the ranges of 112–1,925 aa, 12,509.41–215,896.33 kDa, 4.45–10.26, and 25.51–79.1, respectively. Forty-nine proteins are considered stabilized because their instability indexes are under 40. All proteins are considered hydrophilic because their grand average of hydropathicity values are negative. According to the results of the subcellular localization prediction, 207 proteins, or 68.31% of all proteins, are found in the nucleus; 47 proteins are found in the chloroplasts; 33 proteins are found in the mitochondria; 12.8% are found in the cytoplasm; two are found in the peroxisomes; and two are extracellular. The transmembrane structure results showed that three proteins have a transmembrane structure.

## Classification and phylogenetic analysis of PtAP2/ERF protein

As previously described, each subfamily within the AP2/ERF family exhibits distinct structural domain characteristics. We adopted the classification method of the *AP2/ERF* gene family in *Arabidopsis*, considering their structural domain features and sequence similarities, to systematically classify the *AP2/ERF* gene family in loblolly pine. We constructed a phylogenetic tree based on the AP2 domain sequences of *A. thaliana*, *P. massoniana*, and *P. taeda* (Fig. 1). It can be seen that the majority of the branches in the phylogenetic tree contain members of three species, suggesting that they may share a common origin. Based on the phylogenetic tree and sequence similarity, we found that all AP2/ERF proteins can be divided into 13 groups: the AP2 subfamily, the RAV subfamily, the ERF subfamily (Groups I–X), and Soloist. The AP2 subfamily consists of 34 sequences, including nine sequences with double domains and 25 sequences with single domains. The RAV subfamily comprises 17 sequences, of which 15 possess one AP2 domain and one B3
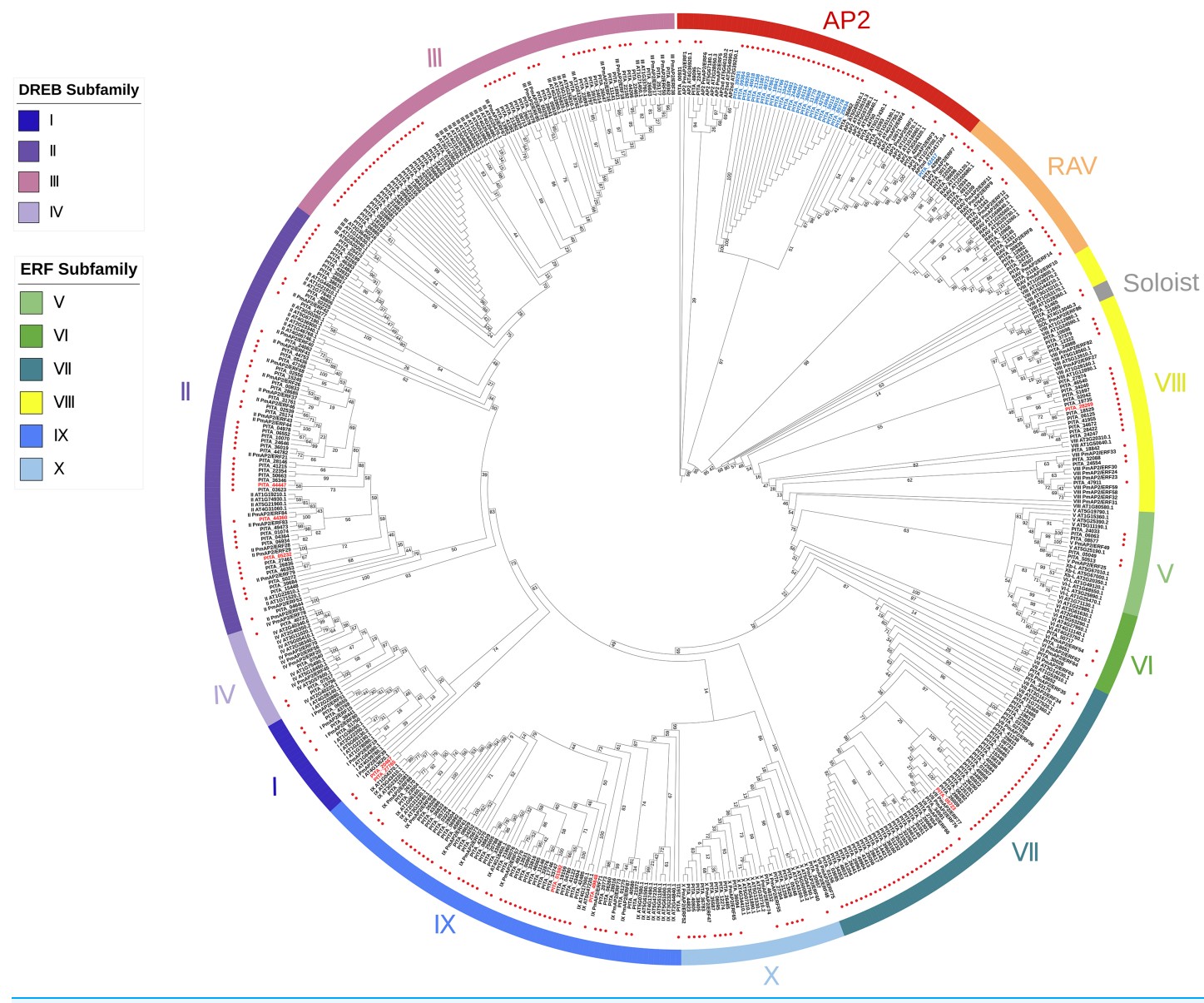

**Figure 1** **A phylogenetic tree based on the AP2 domain sequences of *A. thaliana*, *P. massoniana*, and *P. taeda*.** Red dots represent the proteins in loblolly pine. AP2 subfamily, RAV subfamily, ERF subfamily, and Soloist are grouped and represented by various colored circle strips. Blue labels represent sequences with a single AP2 domain in the AP2 subfamily. Red labels represent sequences with a double AP2 domain in the ERF subfamily.                                      

domain. Only PITA_21543 and PITA_42507 contain one AP2/ERF domain in this family. The ERF family includes Groups I to X, and the ERF family can be further subdivided into the DREB and ERF subfamilies. Groups I to IV belong to the DREB family, with 4, 49, 48, and 4 members, respectively. Among them, three members contain double AP2/ERF domains (PITA_05232, PITA_44360, PITA_44447). Groups V to X belong to the ERF family, with 5, 3, 54, 22, 47, and 15 members, respectively. Among them, six members contain double AP2/ERF domains (PITA_01592, PITA_05723, PITA_20467, PITA_27765, PITA_28269, PITA_49848). Interestingly, PITA_21665 contains three AP2/

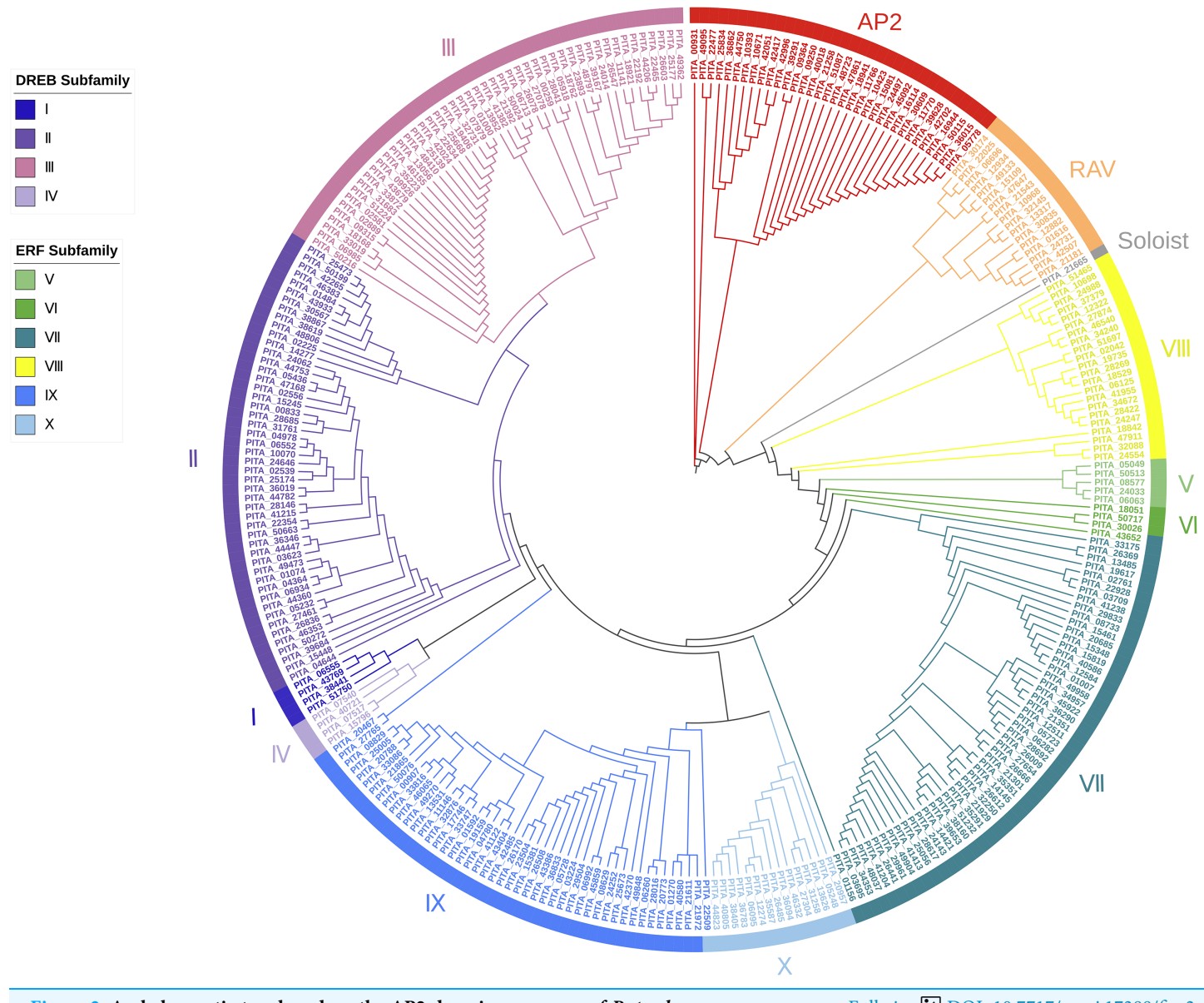

**Figure 2 A phylogenetic tree based on the AP2 domain sequences of *P. taeda*.**

ERF domains and one B3 domain, making it the largest AP2/ERF protein, with a length of 1925 amino acid residues. It cannot be accurately categorized based on its structural domain characteristics as it does not fit the structural domain characteristics of all AP2/ERF gene subfamilies. In the phylogenetic tree, PITA_21665 branches into a clade (Soloist), which includes the members *A. thaliana* and *P. massoniana*. Based on the clustering shown in the phylogenetic tree, it appears to be more similar to sequences classified as Soloists.

Based on the results of the characteristics of domains and the phylogenetic tree, we classified the 303 AP2/ERF superfamily members in loblolly pine into 13 subfamilies (Fig. 2). The specific classification is presented in Table 2.

**Table 2 Classification of the AP2/ERF family in *P. taeda*.**

| Classification | Structural domain features | No. | Total |
|---|---|---|---|
| AP2 family | Double AP2/ERF domain | 9 | 34 |
| | Single AP2/ERF domain | 25 | |
| ERF family (Groups I–IV) | Double AP2/ERF domain | 3 | 105 |
| | Single AP2/ERF domain | 102 | |
| ERF family (Groups V–IX) | Double AP2/ERF domain | 6 | 146 |
| | Single AP2/ERF domain | 140 | |
| RAV family | A AP2/ERF domain and a B3 domain | 15 | 17 |
| | Single AP2/ERF domain | 2 | |
| Soloist | Double AP2/ERF domain and a B3 domain | | 1 |
| | Total | | 303 |

## Conserved domain of analysis of PtAP2/ERF protein

In this study, all sequences containing the complete AP2 domain in loblolly pine were grouped into the AP2/ERF superfamily, and 303 members were analyzed to determine their structural domains. The results of the structural domain analysis showed that the types of domains are mainly the AP2 domain and B3 domain. In total, 303 members of the AP2/ERF superfamily in loblolly pine contain at least one highly conserved AP2 domain (Fig. 3). Generally, the AP2 subfamily comprises double AP2 domains. However, in loblolly pine, the AP2 subfamily includes members with both double and single AP2 domains, which has also been observed in other species, such as *A. thaliana Nakano et al. (2006)*, *Erianthus fulvus Qian et al. (2023)* and *Dendrobium catenatum Han et al. (2022)*. In the ERF and DREB subfamilies, 96.4% of the members contain a single AP2 domain, and nine members (PITA_05232, PITA_44360, PITA_44447, PITA_05723, PITA_28269, PITA_01592, PITA_20467, PITA_27765, and PITA_49848) contain double AP2 domains. The B3 domain is present in the RAV subfamily and is positioned at the 3′ end of the sequence relative to the AP2 domain. Two members (PITA_21543 and PITA_42507) in the RAV subfamily lack the B3 domain. PITA_21665 contains one B3 domain and three AP2 domains. The B3 domain is located at the N-terminus of the sequence in relation to the AP2 domain, differing from the characteristics of the RAV domain.

We performed multiple sequence alignment of the conserved domains within each subfamily in accordance with the classification of the PtAP2/ERF protein (Figs. S1–S3). The AP2 subfamily with double domains consists of nine members, and their domain sequences are relatively conserved and interconnected by a conserved sequence of 30 amino acids. They contain an intact YRG element and intact RAYD elements, almost all of which include the YLG motif (Fig. S1). The members of the AP2 subfamily with a single domain show a higher degree of sequence conservation among each other, but this sequence conservation differs from that in members with two domains (Fig. S1). The AP2/ERF domains within the RAV subfamily exhibit a high degree of conservation, all of which include WLG motifs. However, it should be noted that the AP2/ERF domains of

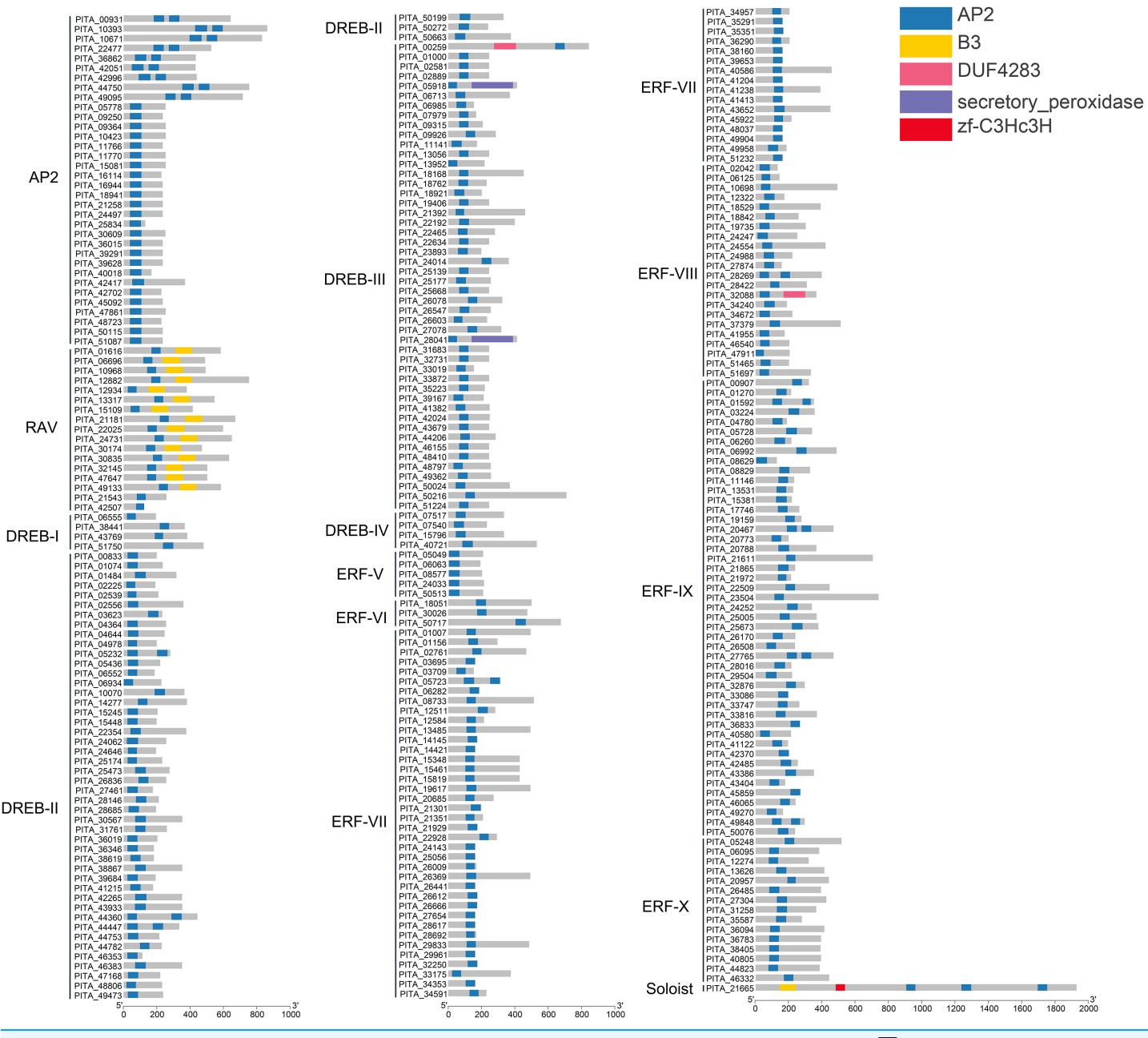

**Figure 3** Conserved domain analysis of PtAP2/ERF TFs.

PITA_42507 show a C-terminal deletion. Despite the absence of the B3 domain in PITA_21543 and PITA_42507, they are classified within the RAV subfamily due to the substantial homology between their AP2/ERF domains (Fig. S1). The ERF family is further divided into the DREB and ERF subfamilies based on the amino acid residues at positions 14 and 19 (*Nakano et al., 2006*). In the DREB subfamily, the amino acids at positions 14 and 19 of the AP2/ERF domain are valine (V) and glutamic acid (E), respectively. In the ERF subfamily, the amino acid residues at positions 14 and 19 of the AP2 domain are alanine (A) and aspartic acid (D), respectively. In the DREB family of loblolly pine, the

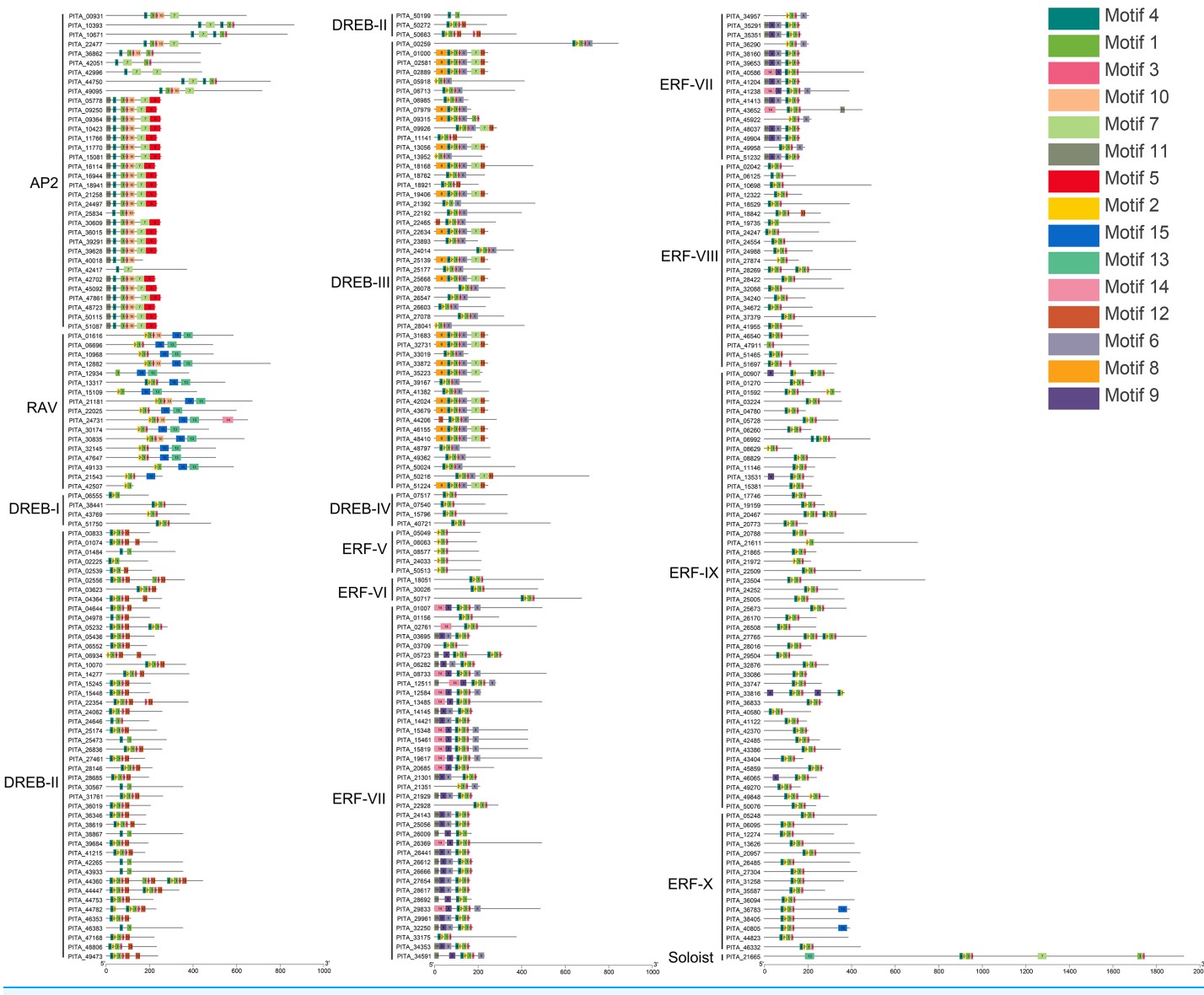

**Figure 4 Conserved motif analysis of PtAP2/ERF TFs.**

majority of sequences have valine (V) at position 14, while a few sequences show variation in the amino acid at position 14, indicating a lack of conservation (Fig. S2). In the ERF family, the majority of sequences have alanine (A) at position 14, with a few sequences showing variation. Moreover, more than 90% of the sequences in the ERF family have aspartic acid (D) at position 19 (Fig. S3). Most sequences in both the DREB and ERF families contain the WLG motif, while a few contain the SLG, HLG, or CLG motifs. The results of the multiple sequence alignment revealed specific conserved residues within the AP2 domains of each subgroup, consistent with the patterns depicted in the phylogenetic tree.
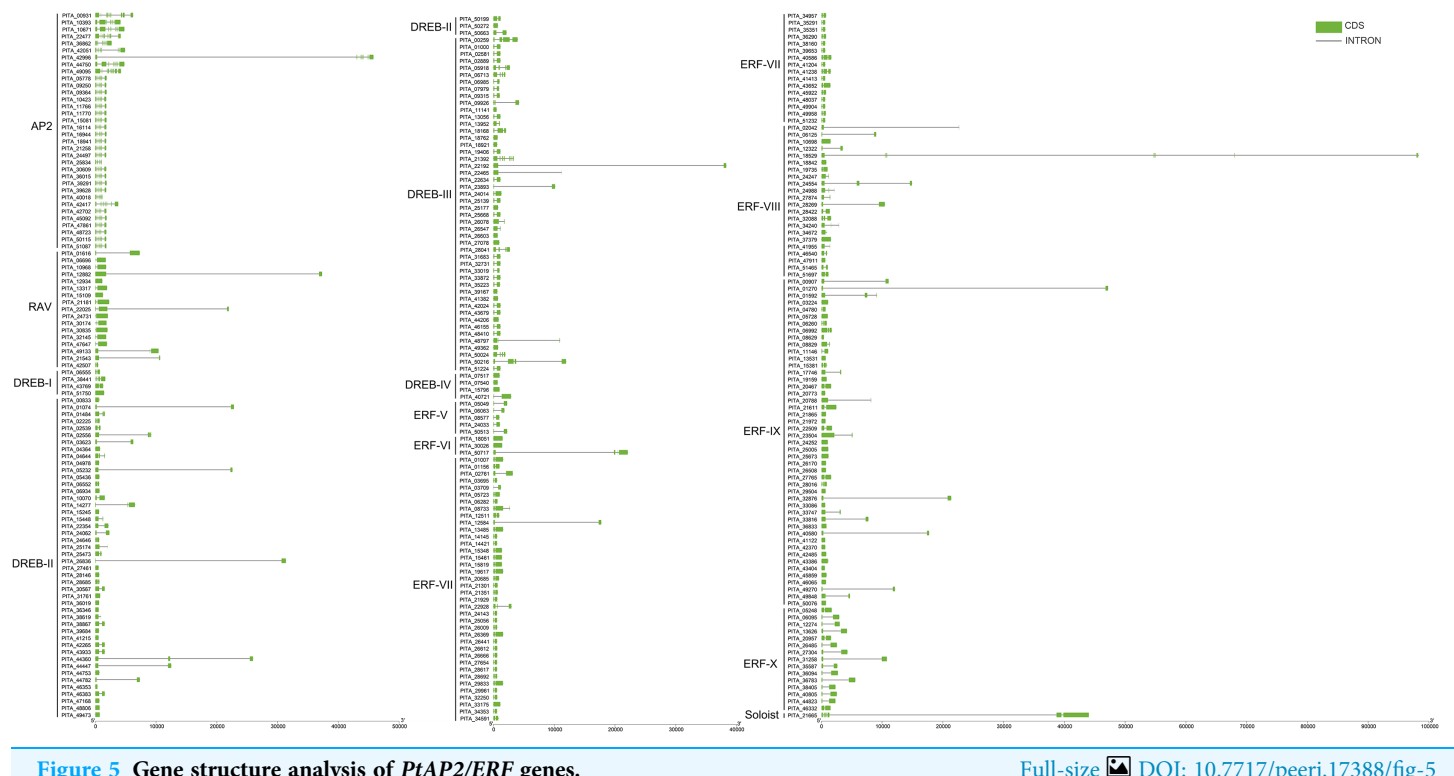

**Figure 5  Gene structure analysis of *PtAP2/ERF* genes.**     

## Conserved motif and gene structure analysis of *PtAP2/ERF* TFs

Conserved motif analysis of the AP2/ERF family in loblolly pine was performed using the online MEME program and visualized with TBtools (Fig. 4), and a total of 15 conserved motifs were detected. The conserved motifs in the domains were analyzed in conjunction with Figs. 3 and 4. Motifs 1, 2, 3, 4, and 7 are sequences that make up the AP2 domains, and Motifs 13 and 15 are the basic sequences that make up the B3 domains. The types of motifs in the conserved structural domains vary across subfamilies, *e.g.*, the AP2 domain in the AP2 subfamily contains Motifs 1, 3, 4, and 7; the AP2 domain of the RAV subfamily contains Motifs 1, 2, and 3, and its B3 domain contains Motifs 13 and 15; the AP2 domains of the ERF and DREB subfamilies contain Motifs 1, 2, 3, and 4; and each of the three AP2 domains of *PITA_21665* contains different motifs (the first AP2 domain contain Motifs 1, 2, and 3, the second contains Motif 7, and the third contains Motifs 1 and 3); and the B3 domain contains Motif 15. The AP2 structural domain and its conserved motifs exhibit greater similarity within the ERF and DREB subfamilies. Upon analyzing the conserved motifs outside the structural domains, we found that the same subfamilies contain similar motifs, and different subfamily members contain their own specific conserved motifs; for example, Motif 5 presents only in the AP2 subfamily, Motifs 8 and 12 only in the DREB subfamily, and Motifs 9 and 14 only in the ERF subfamily.

To further understand the characterization of PtAP2/ERF TFs, the gene structures were visualized using TBtools software (Fig. 5). The results showed similar gene structures within a subfamily. Members of the *AP2* subfamily all have introns, which are more numerous compared to those of other subfamilies, with as many as nine and as few as

three. Compared with the *AP2* subfamily, the *RAV*, *ERF*, and *DREB* subfamilies contain relatively few introns, usually around zero to three. Among them, 63 genes contain no introns, 168 genes contain one intron, 25 genes contain two introns, and 10 genes contain three introns. *PITA_18529* in ERF VIII has six introns, and *PITA_21392* in DREB III contains seven introns. *PITA_21665* in SOL has five introns.

Through an analysis of the conserved domains, conserved motifs, and gene structures of the loblolly pine AP2/ERF superfamily, we unveiled pronounced characteristic disparities between distinct subfamilies. Concurrently, each subfamily internally exhibits analogous traits. On one hand, the presence of conserved domains and motifs signifies that these genes have been subjected to a process of conservative selection over the course of evolution. This phenomenon is potentially associated with their vital roles in fundamental biological processes, such as plant growth and responses to environmental changes. Conversely, the observed discrepancies between different subfamilies imply that these genes have potentially undergone differentiation in specific biological functions and regulatory pathways to accommodate diverse ecological and physiological demands.

## Cis-acting element analysis of *PtAP2/ERF* gene promoters

This analysis revealed 69 types of cis-acting elements within the promoter regions of *PtAP2/ERF* genes. These elements were identified after excluding basal elements (TATA-box and CAAT-box) and those with unknown functions. Among them, 35 types are associated with the light response, 16 with the phytohormone response, nine with the stress response, and nine with plant growth and development (Fig. 6; Table S3). The light-responsive elements are the most numerous in terms of type and quantity. Most elements are associated with the part of a conserved DNA module involved in light responsiveness or a light-responsive element. There is a presumption that the *PtAP2/ERF* gene is responsible for the light signal response. Phytohormone-responsive elements are abundant, such as the MeJA-responsive element (926), abscisic acid-responsive element (762), ethylene-responsive element (508), salicylic acid-responsive element (311), gibberellin-responsive element (279), and auxin-responsive element (93). Stress induction elements are associated with anaerobic induction (821), the damage-responsive element (293), drought inducibility (228), the wound-responsive element (163), low-temperature responsiveness (127), and other factors. The DRE element, which is involved in the response to dehydration, low-temperature, and salt stress, is a cis-acting element that functions in the promoter regions of its target genes and interacts with DREB TFs. The complete DRE element is present in the promoter regions of *PITA_25673* and *PITA_12274*, and there is a presumed interaction between AP2/ERF TFs (Table S1). Fewer elements related to plant growth and development are associated with meristem expression (128), zein metabolism regulation (127), endosperm expression (70), circadian control (60), and other factors. These results suggest that *PtAP2/ERF* genes may play a significant part in photosynthesis, hormone signaling pathways, the biotic and abiotic stress response, and plant growth and development, and that there may be time- and tissue-specificity in *PtAP2/ERF* gene expression.

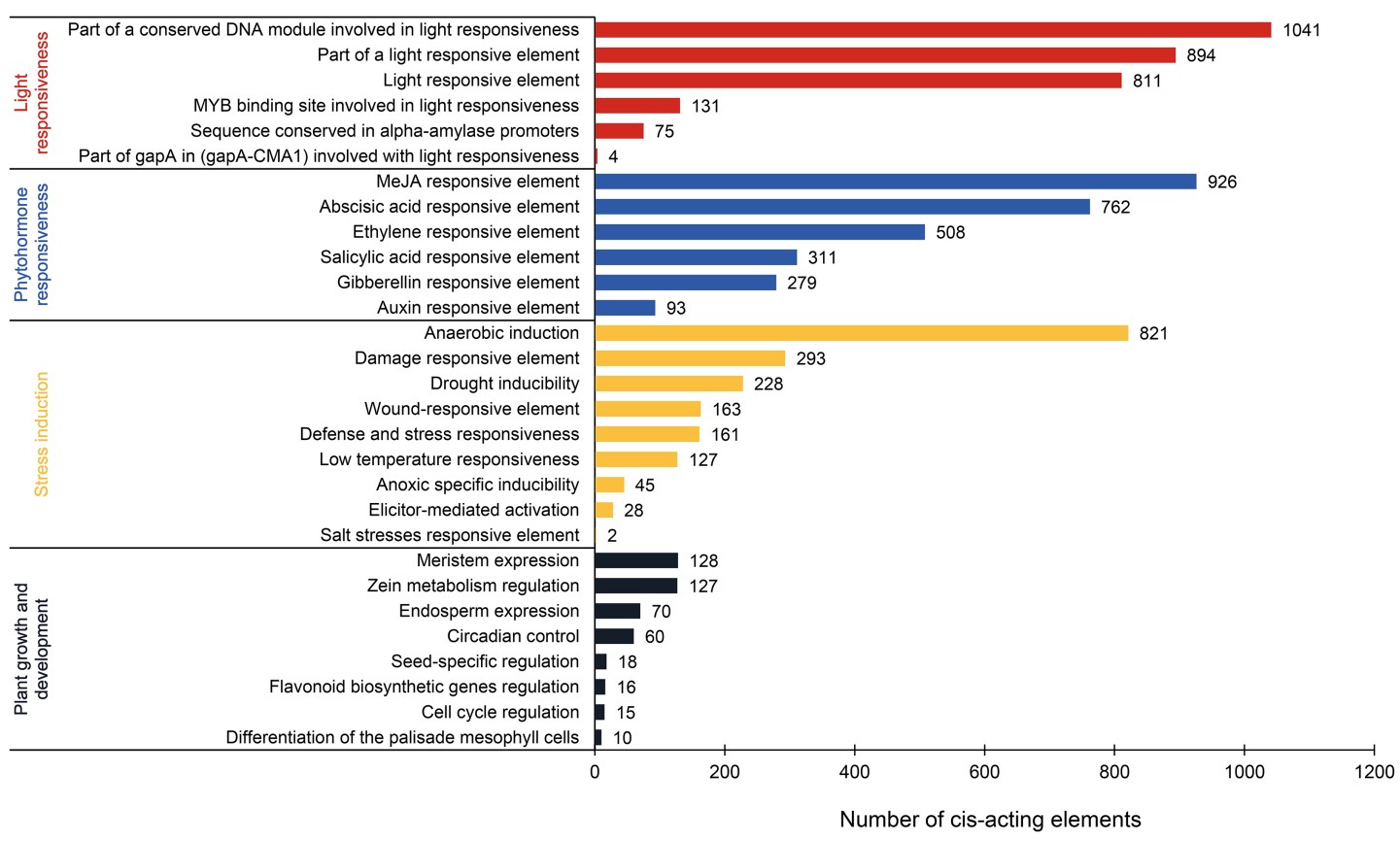

**Figure 6  Analysis of cis-acting elements in the PtAP2/ERF promoters of loblolly pine.**

## Prediction and analysis of PtAP2/ERF target genes

Previous studies have indicated that the DRE/CRT and GCC-box elements serve as specific binding sites for AP2/ERF transcriptional regulation of downstream target genes (*Ohme-Takagi & Shinshi, 1995*; *Sakuma et al., 2002*). We considered genes that contain the DRE/CRT element or GCC-box element in their promoter regions as potential target genes of PtAP2/ERF TFs. Our search of the loblolly pine genome revealed 8,952 genes with at least one DRE/CRT element in their promoters, and 1,068 genes with at least one GCC-box element. Taking the union of these two gene sets, a total of 9,740 potential PtAP2/ERF target genes were confirmed and used in the subsequent GO and KEGG analysis.

The GO annotation results showed that 4,862 genes were annotated and localized to 57 functional groups, including 25 for biological processes, 18 for cellular components, and 14 for molecular function ($p < 0.05$). In terms of biological processes, these target genes are mainly focused on cellular processes, metabolic processes, single biological processes, biological regulation, and plant growth and development; in terms of cellular composition, they are mainly associated with cell parts such as cell membranes and organelles; and in terms of molecular functions, they are mainly focused on catalytic activity and binding effects (Fig. 7A).

The KEGG annotation results showed that 4,888 genes were annotated. The target genes were annotated to various metabolic pathways, genetic information transfer,

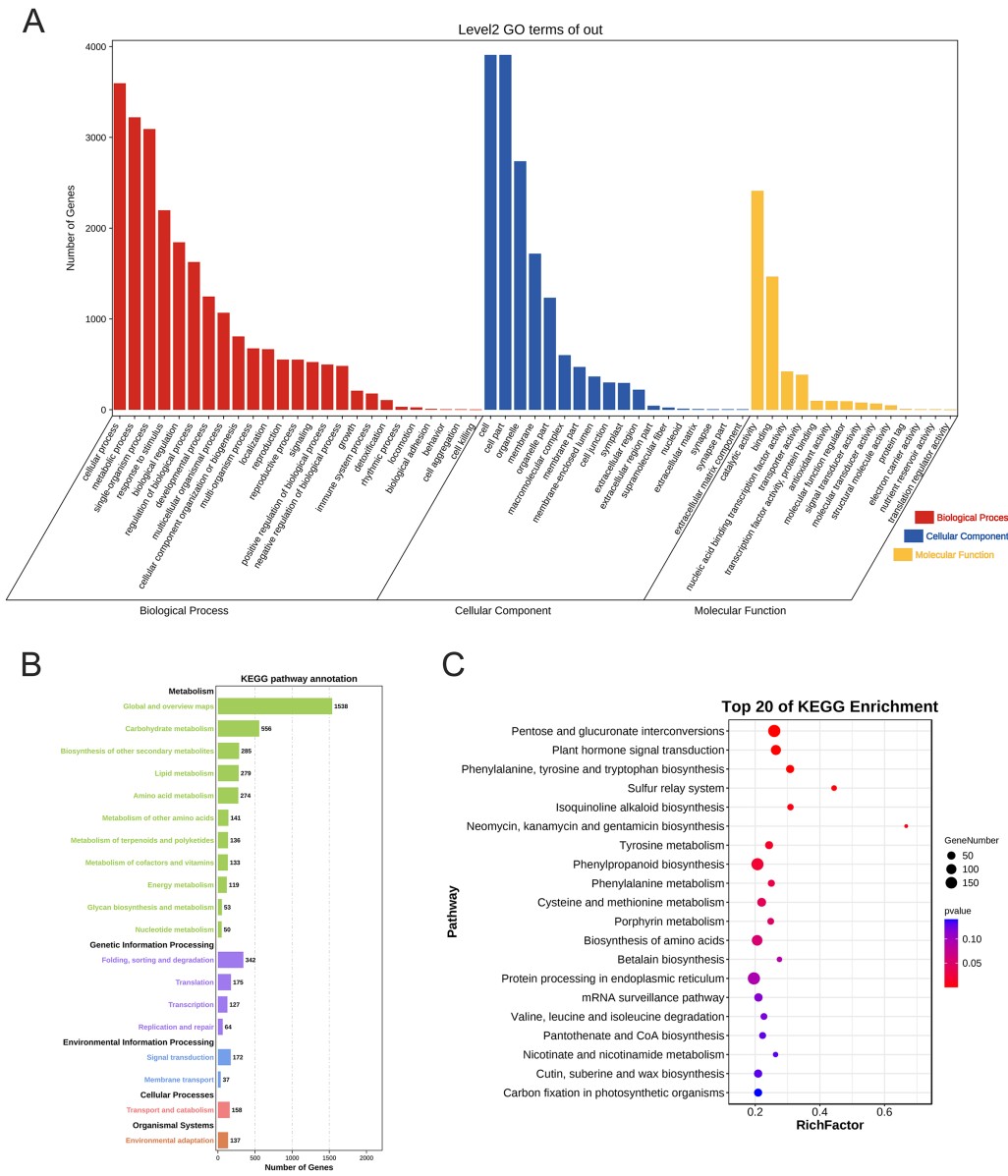

**Figure 7 Gene Ontology annotation and KEGG enrichment analysis of PtAP2/ERF target genes.** (A) GO annotation of PtAP2/ERF predicted target genes. (B) KEGG pathway annotation of PtAP2/ERF predicted target genes. (C) Top 20 of KEGG enrichment of PtAP2/ERF predicted target genes.

environmental information processing, cellular processes, and organismal systems ($p < 0.05$) (Fig. 7B). We conducted hypergeometric tests at the level of KEGG Pathways to identify pathways that are significantly enriched in PtAP2/ERF potential target genes compared to the entire genomic background, with a significance threshold set at $p < 0.05$. The top 20 of KEGG enrichment analysis showed that the pathways were mainly enriched in pentose and glucuronate interconversions ($p < 0.01$), plant hormone signal transduction ($p < 0.01$), phenylpropanoid biosynthesis ($p < 0.01$), cysteine and methionine metabolism ($p < 0.05$), the biosynthesis of amino acids ($p < 0.05$), *etc.* (Fig. 7C).

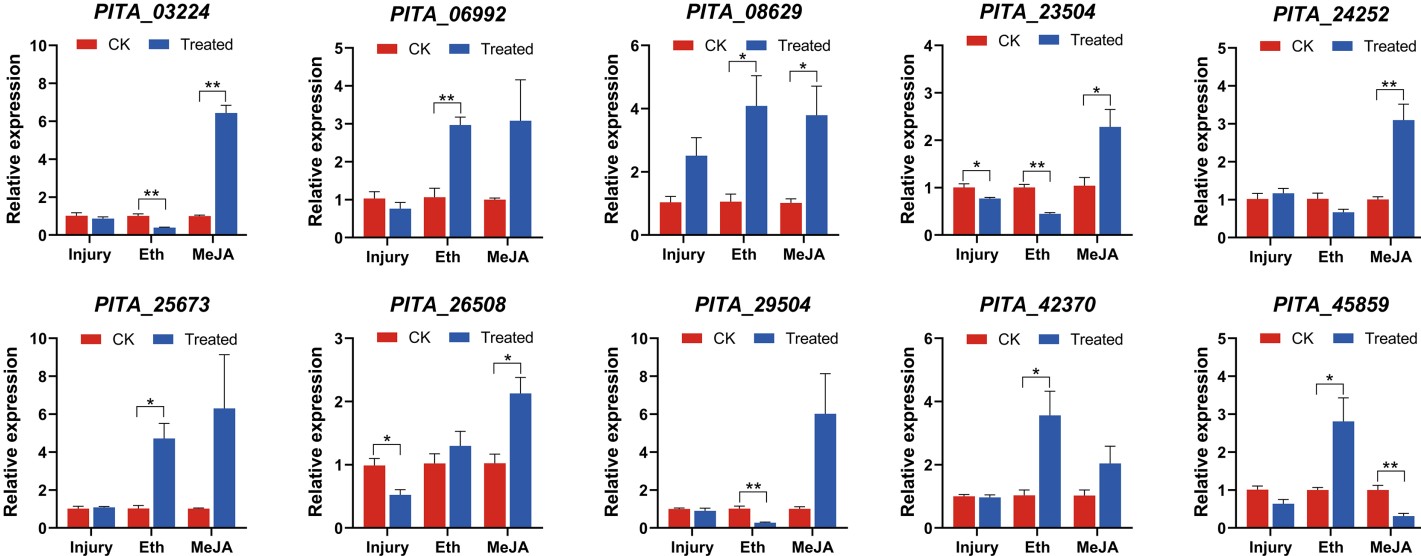

**Figure 8 qRT-PCR expression analysis of ten selected *PtAP2/ERF* genes in Group IX under mechanical injury, ethephon (Eth), and methyl jasmonate (MeJA).** The mean from three different experiments was computed for each column. The standard error of the mean is represented by error bars. There were significant differences between controls and experimental subjects (One-way ANOVA) as indicated by $^*p < 0.05$ and $^{**}p < 0.01$.

## The expression of *PtAP2/ERFs* in group IX under defense-related treatments

The transcript profiles in the needles were assessed *via* qRT-PCR 6 h after the initiation of treatment and were subsequently compared to the control samples. Overall, we observed differential expression patterns among the ten genes under the three different treatments (Fig. 8). Most *ERF* genes did not exhibit significant changes in their expression under mechanical injury treatment, but their expression was significantly altered under the Eth and MeJA treatments. Eth and MeJA are effective components of resin-promoting stimulants. The differential expression patterns of genes between mechanical injury and hormone treatments reflect that these genes are more likely to be induced by hormone treatments, aligning with the principles of applying stimulants.

Under the mechanical injury experiment, *PITA_26508* and *PITA_23504* were significantly downregulated 0.52-fold and 0.77-fold compared with the CK1 ($p < 0.05$), respectively. Other genes might not exhibit a responsive reaction to mechanical damage.

Under the Eth treatment experiment, five genes (*PITA_06992*, *PITA_08629*, *PITA_25673*, *PITA_42370*, *PITA_45859*) were significantly upregulated, with fold changes of 2.97, 4.09, 4.72, 3.56, and 2.81, respectively, compared to the control group ($p < 0.05$). In contrast, *PITA_03224*, *PITA_23504*, and *PITA_29504* were significantly downregulated 0.40-fold, 0.45-fold and 0.28-fold ($p < 0.01$), respectively.

Nine genes exhibited induced upregulation under MeJA treatment, with the following fold changes: *PITA_03224* (6.44-fold), *PITA_06992* (3.08-fold), *PITA_08629* (3.79-fold), *PITA_23504* (2.28-fold), *PITA_24252* (3.10-fold), *PITA_25673* (6.31-fold), *PITA_26508* (2.13-fold), *PITA_29504* (6.02-fold), and *PITA_42370* (2.04-fold). Among them, except for

*PITA_06992*, *PITA_25673*, and *PITA_29504*, all genes exhibited significant upregulation ($p < 0.05$). *PITA_45859* decreased 0.32-fold ($p < 0.01$).

These ten genes exhibited two distinct expression patterns under the hormonal treatments. The first pattern consists of upregulation under both the Eth and MeJA treatments, such as in *PITA_06992*, *PITA_08629*, *PITA_25673*, and *PITA_42370*. Among them, *PITA_08629* exhibited significant upregulation ($p < 0.05$). This indicates that *PITA_08629* is jointly induced by Eth and MeJA and may play a regulatory role in resin biosynthesis. The second pattern involves upregulation after MeJA treatment but a decrease in expression after the Eth treatment, such as in *PITA_03224*, *PITA_23504*, *PITA_24252*, and *PITA_29504*. Among them, *PITA_03224* and *PITA_23504* exhibited significant differential expression ($p < 0.05$). Several members of the same *ERF* subgroup could be induced by different hormones, suggesting that plant responses to stress are diverse.

## DISCUSSION

The AP2/ERF TF family is abundant in the plant kingdom, playing roles in plant growth and development, the response to biotic and abiotic stresses, and metabolite biosynthesis, and is a very important class of TFs. Due to the functional diversity of the AP2/ERF TF family, genome-wide identification, classification, and functional analysis of the *AP2/ERF* gene family have been carried out in a variety of plants. However, due to the large genome size of Pinus species, such as *Pinus lambertiana* (27.6 G) *Crepeau, Langley & Stevens (2017)*, *Pinus tabuliformis* (25.4 G) *Niu et al. (2022)*, and *P. taeda* (22.1 G) *Zimin et al. (2017)*, genome assembly and annotation are challenging, and genome-wide characterization of the *AP2/ERF* gene family has been performed less frequently in Pinus species. Genome-based investigations provide a more comprehensive approach compared to gene family analyses conducted on transcriptome or EST data. Therefore, the *AP2/ERF* gene family of loblolly pine was identified and analyzed genome-wide. In addition, we explored the expression of ten members in group IX under three defense-related treatments (mechanical injury, Eth, and MeJA) *via* qRT-PCR.

There are 303 genes in the *PtAP2/ERF* gene family, which is more than the number of *AP2/ERF* gene family in most other species. Similar to many other species, the quantity of the *PtAP2/ERF* gene family depends on how many members there are in the ERF subfamily (Table 2). Across plant species, the percentage of genes in each family relative to the total *AP2/ERF* superfamily genes is similar, indicating a certain degree of conservation during gene family evolution. Previous studies have indicated that the number of *AP2/ERF* gene family members is not necessarily connected with genome size (*Sun et al., 2022*). However, 70–80% of the conifer's oversized genome is represented by repetitive sequences (*Raj & Neale, 2005*), and we hypothesize that this may also be a factor influencing the number of *AP2/ERF* gene families. The wide growth range and strong adaptability of loblolly pine might contribute to the higher number of *AP2/ERF* gene family members it possesses, as this gene family is widely involved in the stress responses (*Niu et al., 2022*; *Lu, Krutovsky & Loopstra, 2019*).

It is known that different subgroups within the *AP2/ERF* gene family possess distinct functions. Therefore, the classification of gene families can provide a foundation for subsequent functional investigations. To precisely classify the *PtAP2/ERF* gene family, we constructed a phylogenetic tree of the AP2/ERF TFs' AP2 domain sequences from the model plant *A. thaliana*, the closely related species *P. massoniana*, and loblolly pine. The classification was informed by the results of the multiple sequence alignments of the *PtAP2/ERF* gene family. In the evolutionary tree, the subfamilies of loblolly pine have similar clustering and distribution to those of *A. thaliana* and *P. massoniana* (Fig. 1). The results from the multiple sequence alignment revealed that the fourteenth amino acid of the ERF-V members (PITA_05049, PITA_06063, PITA_08577, PITA_24033, PITA_50513) in loblolly pine is valine (V), and the nineteenth amino acid is histidine (H), consistent with the findings in slash pine. This consistency reflects the structural similarities among species with closer phylogenetic relationships (Fig. S3) (*Sun et al., 2022*). However, based on the findings of the phylogenetic tree, loblolly pine's ERF-V is a branch within the ERF subfamily; thus, we continue to consider this group to be a subgroup of ERF rather than DREB.

The conserved domain and motif analyses revealed that 74% of the AP2 subfamily members contain only one AP2 domain, and two RAV subfamily members (PITA_21543, PITA_42507) lack a complete B3 domain. This absence of domains is inferred to result from gene mutations caused by evolutionary variations (Figs. 3, 4). The gene structure analysis indicated that the AP2 subfamily members generally possess a higher number of introns (3–9) compared to the RAV and ERF subfamilies (0–3) (Fig. 5). Studies have suggested that introns may gradually disappear in evolution (*Roy & Penny, 2007*). Therefore, we postulate that the RAV and ERF subfamilies might have evolved from the AP2 subfamily. In summary, members within the same subfamily exhibit similar domain compositions and distributions of conserved motifs, and comparable gene structures, while variations exist between different subfamilies, corroborating the results of the phylogenetic tree classification.

To gain preliminary insights into the functional roles of the *AP2/ERF* gene family in loblolly pine's physiological activity, we analyzed the cis-regulatory elements and predicted target genes. Similar to *D. catenatum Han et al. (2022)* and *Boehmeria nivea Qiu et al. (2022)*, the results of the cis-regulatory element analysis revealed the presence of numerous light-responsive elements, hormone-responsive elements, and stress-responsive elements (Fig. 6). Light-responsive elements are widely distributed in the promoters of light-regulated genes. The promoter region of the *PtAP2/ERF* gene family encompasses a substantial number of light-responsive elements, indicating their nature as light-regulated genes. The expressed protein in this context might function as a transcription factor involved in light signal transduction. Plants regulate their physiological status based on environmental changes during growth and development. Plants regulate various developmental processes, including seed germination, seedling photomorphogenesis, shade avoidance, circadian rhythms, flower induction, and senescence, in response to changes in light signals (*Jing & Lin, 2020*). The rich presence of hormone-responsive elements and stress-responsive elements in the promoter region of the *PtAP2/ERF* gene

family suggests their potential involvement in a broad range of responses to environmental stress. For instance, PtoERF15 has been shown to enhance drought tolerance through jasmonate-mediated signaling in *Populus tomentosa* *Kong et al. (2023)*. Our related analysis of target genes indicated that the regulatory pathways of the *AP2/ERF* gene family primarily focus on plant hormone signaling transduction, as well as the biosynthesis of amino acids or proteins (Fig. 7). During plant growth and development, changes in phytohormone levels occur as an initial response to external environmental stress. This change manifests in the management of abiotic stress through aspects such as osmotic stress sensing and signaling, gene regulation under abiotic stress, growth regulation under abiotic stress, and the regulation of stomatal movement (*Waadt et al., 2022*). Combining both analyses, it is hypothesized that the *AP2/ERF* gene family may be involved in plant growth and development and the stress response through responses to light signals, hormone-induced signaling pathways, or the direct regulation of related gene expression.

Stresses such as mechanical damage, insect feeding, and exogenous compounds such as Eth and MeJA induce traumatic resin tracts in conifers, and several cascading defense responses occur, such as the activation of polyphenolic thin-walled cells and the accumulation of resin (*Franceschi et al., 2005*; *Vázquez-González et al., 2022*; *Mason et al., 2019*). This is an important component of the defense response of conifers. Members of Group IX have been implicated in defense responses and secondary metabolite biosynthesis in earlier research (*Wang et al., 2022*; *Srivastava & Kumar, 2019*). It has been shown that the application of stimulants containing Eth or MeJA can significantly increase resin production (*Lopez-Alvarez et al., 2023*; *de Oliveira Junkes et al., 2019*). Therefore, we examined the relative expression of 10 genes of Group IX under three treatments (injury, Eth, and MeJA) *via* qPCR. The expression of the ten *ERF* family genes differed under the injury, ETH, and MeJA treatments, suggesting that the *AP2/ERF* genes may be functionally diverse among treatments, and may go through different signaling pathways to respond to stress and regulate resin biosynthesis. MeJA can improve plant stress tolerance, such as resistance to cold (*Chen et al., 2021*), heat (*Liao et al., 2023*), drought (*Javadipour et al., 2019*), pests and diseases (*Krokene et al., 2023*), *etc.* Under the MeJA treatment experiment, it was observed that the expression of the majority of the detected *AP2/ERF* genes was upregulated. These identified *AP2/ERF* genes in loblolly pine might be involved in various stress responses through the MeJA signaling pathway. Research has indicated that the overexpression of specific *AP2/ERF* genes can enhance plant resistance and terpenoid production. For instance, transgenic tobacco's tolerance to the waterlogging stress was greatly increased by the heterologous expression of *PjERF13* (*He et al., 2023*); resistance to drought stress was enhanced by the overexpression of *AtruDREB28* (*Li et al., 2023*); and the generation of monoterpenes in *Litsea cubeba* was considerably boosted by the overexpression of *LcERF134* (*Zhao et al., 2023*). Exploring key *PtAP2/ERF* genes that respond to stress holds the potential to enhance its resistance and broaden its suitable growth range. Furthermore, investigating the crucial *PtAP2/ERF* genes involved in loblolly pine's resin biosynthesis pathways might offer novel insights from a genetic improvement perspective, and could be useful for increasing resin yield. Our KEGG pathway analysis of PtAP2/ERF TF target genes revealed enrichment in the hormone signaling pathways and

protein synthesis pathways (Fig. 7). The results of RT-qPCR demonstrated distinct expression patterns among some members of Group IX under three defense-related treatments (Fig. 8). These findings provide a theoretical foundation for enhancing resin production and strengthening resistance at the transcriptional regulation level for loblolly pine. However, achieving this goal requires further functional analysis of PtAP2/ERFs and an exploration of their regulatory mechanisms with downstream target genes.

The loblolly pine genome was not assembled at the chromosome level, which imposes certain limitations on conducting gene family analysis in this study. We are unable to determine the chromosomal location of *PtAP2/ERF* genes, potentially leading to instances of gene duplication or omission in gene family analysis. Moreover, it may also constrain a comprehensive understanding of *PtAP2/ERF* gene functions, as functional annotation sometimes requires the interpretation of gene location on chromosomes in conjunction with complete genome information. With the continuous development of sequencing technologies, we believe that these issues are expected to be addressed in the future.

In conclusion, our identification, classification, and expression analyses of *PtAP2/ERF* family genes provide an initial understanding of the biological functions of AP2/ERF TFs in loblolly pine. However, further research is needed to elucidate the regulatory mechanisms through which PtAP2/ERF TFs participate in resin synthesis. Additionally, investigating the involvement of PtAP2/ERF TFs under different stress conditions remains a topic worthy of investigation.

## CONCLUSIONS

In this study, the identification and classification of the AP2/ERF family in loblolly pine were conducted, and the expression patterns of selected Group IX members under three defense-related treatments (mechanical injury, Eth, MeJA) were preliminarily explored. This study of the PtAP2/ERF family is the first genome-wide investigation into the AP2/ERF family in Pinaceae species. Our results indicated that certain members of the PtAP2/ERF family in group IX exhibited different responses under mechanical injury, Eth, and MeJA treatments, suggesting that these members may participate in stress responses through hormone signal transduction. In conclusion, our study lays the groundwork for future functional analyses of the PtAP2/ERF superfamily and serves as a guide for identifying further Pinaceae species.

## ACKNOWLEDGEMENTS

We thank the Yingde Research Institute of Forestry in Guangdong.

### Funding

This study was funded by the Key-Area Research and Development Program of Guangdong Province (No. 2020B020215001), and the Guangdong Basic and Applied Basic Research Foundation (2021A1515011098). The funders had no role in study design, data collection and analysis, decision to publish, or preparation of the manuscript.

### Grant Disclosures
The following grant information was disclosed by the authors:
Key-Area Research and Development Program of Guangdong Province: 2020B020215001.
Guangdong Basic and Applied Basic Research Foundation: 2021A1515011098.

### Competing Interests
The authors declare that they have no competing interests.

### Author Contributions
- Peiqi Ye performed the experiments, analyzed the data, prepared figures and/or tables, authored or reviewed drafts of the article, and approved the final draft.
- Xiaoliang Che performed the experiments, authored or reviewed drafts of the article, and approved the final draft.
- Yang Liu analyzed the data, authored or reviewed drafts of the article, and approved the final draft.
- Ming Zeng performed the experiments, authored or reviewed drafts of the article, and approved the final draft.
- Wenbing Guo analyzed the data, authored or reviewed drafts of the article, and approved the final draft.
- Yongbin Long analyzed the data, authored or reviewed drafts of the article, and approved the final draft.
- Tianyi Liu conceived and designed the experiments, authored or reviewed drafts of the article, and approved the final draft.
- Zhe Wang conceived and designed the experiments, authored or reviewed drafts of the article, and approved the final draft.

### Data Availability
The raw data of gene relative expression by qRT-PCR is available in the Supplemental File.

### Supplemental Information
Supplemental information for this article can be found online at http://dx.doi.org/10.7717/peerj.17388#supplemental-information.

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
