# Peer review of "Genome-wide identification and characterization of the AP2/ERF gene family in loblolly pine (Pinus taeda L.)"

_PeerJ, doi:10.7717/peerj.17388_

## Round 0.1 · original submission · Minor Revisions

· Academic Editor

Minor Revisions

This is an interesting manuscript on the characterization of the AP2/ERF gene family in loblolly pine, a species with a large complex genome. Two reviewers agreed that minor revisions are needed but that the study is well-conducted and detailed.

I request that the authors consider each of the reviewers' comments and address these. In particular, more details are needed in the methods section on the treatment protocols applied. I agree with one of the reviewers that the gene expression quantification data should be provided in the supplementary information or made available publicly online for transparency.

Reviewer 1 ·

Basic reporting

1)The Latin name of Loblolly pine only needs to be marked in the title, abstract and text where it first appears.
2)Line 26-29“ the AP2/ERF gene family was identified in the whole loblolly pine genome using bioinformatics methods and by systematically analyzing its gene characteristics, physicochemical properties, phylogenetic evolution, promoter cis-elements, expression pattern, and potential functions. ” should be deleted. The sentence is repeated with the subsequent content of the abstract.

Experimental design

1)Line 194, the mechanical injury was not clearly discribed the specific method.
2)Line 189-Line 205, It was not clear how many seedlings were treated respectively, including 3 controls and 3 treatments.

Validity of the findings

There are no direct evidence shows that the conclusion of Line 558 "these members may be involved in the regulation of resin biosynthesis".

Additional comments

All of the Latin name of plants should be italics.

Reviewer 2 ·

Basic reporting

This article presents a detailed survey of AP2/ERF gene family in loblolly pine. Writing is clear, figures and tables are straightforward. However, I did not find the raw data of those ten gene expression quantifications. I think those data should be publicly shared.

Experimental design

Some methods require more description.

1. I feel more description is needed on treatment protocol. For instance, how did you perform “mechanical injury” treatment? Also, please simply explain the function of “Eth” and “MeJA”, I suppose they are provoking resin production. How did you implement such “treatment”? Did you inject or spread the chemical solution? Please clarify.

2. Line385-Line389, to prove the association between DRE/CRT/GCC and PtAP2/ERF TFs, a statistic test, such as hypergeometric test, is better than just showing numbers.

3. Line390-Line403, what statistical methods and threshold P-values were used for GO/KEGG Enrichment analyses? Details like these can help readers assess the importance of the identified terms.

4. Line415-Line433, statistical threshold is necessary to describe these gene expression profiles.

Validity of the findings

A “limitation and shortcomings” part should be added in "Discussion". As far as I know, scaffolds of loblolly pine have not been assembled to chromosomes. Does it affect the results of your gene family/gene element survey?

Additional comments

Typos such as Line79, “n”; Line82, typo “I” ......

---

## Round 0.2 · accepted · Accept

· Academic Editor

Accept

Both reviewers have agreed that their comments and concerns from the first revisions have been addressed and no further concerns were raised. Thank you to the reviewers and authors for your thorough reviews and edits, I recommend this manuscript for acceptance.

Reviewer 1 ·

Basic reporting

The manuscript has been revised according to the reviewer's suggestion.

Experimental design

The manuscript has been revised according to the reviewer's suggestion.

Validity of the findings

The manuscript has been revised according to the reviewer's suggestion.

Additional comments

The manuscript has been revised according to the reviewer's suggestion.

Reviewer 2 ·

Basic reporting

In this resubmission, authors addressed my concerns in the first submission. I don't have further questions.

Experimental design

Authors added details about the "injury method" and qPCR quantification result table, which addressed my concerns in the first submission.

Validity of the findings

No comment